# Economic Injury Levels and Economic Thresholds for *Leucoptera coffeella* as a Function of Insecticide Application Technology in Organic and Conventional Coffee (*Coffea arabica*), Farms

**DOI:** 10.3390/plants13050585

**Published:** 2024-02-21

**Authors:** Marcelo Coutinho Picanço Filho, Eraldo Lima, Daiane das Graças do Carmo, Angelo Pallini, Adriana Helena Walerius, Ricardo Siqueira da Silva, Letícia Caroline da Silva Sant’Ana, Pedro Henrique Queiroz Lopes, Marcelo Coutinho Picanço

**Affiliations:** 1Department of Entomology, Federal University of Viçosa, Viçosa 36570-900, Brazil; marcelo.filho@ufv.br (M.C.P.F.); eraldo.lima@ufv.br (E.L.); pallini@ufv.br (A.P.); adrianahelenawalerius@gmail.com (A.H.W.); pedro.lopes2@ufv.br (P.H.Q.L.); picanco@ufv.br (M.C.P.); 2Department of Agronomy, Federal University of Viçosa, Viçosa 36570-900, Brazil; leticia.ana@ufv.br; 3Department of Agronomy, Universidade Federal dos Vales do Jequitinhonha e Mucuri, Diamantina 39100-000, Brazil; ricardo.ufvjm@gmail.com

**Keywords:** coffee leaf miner, decision making, manual application, tractor application, airplane spraying, drone spraying

## Abstract

*Leucoptera coffeella* (Lepidoptera: Lyonetiidae) is one of the main pests in coffee crops. The economic injury level (EIL) is the lowest density of the pest at which economic damages match the costs of control measures. The economic threshold (ET) is the density of the pest at which control measures must be taken so that this population does not reach the EIL. These are the main indices used for pest control decision-making. Control of *L. coffeella* is carried out by manual, tractor, airplane or drone applications. This work aimed to determine EILs and ETs for *L. coffeella* as a function of insecticide application technology in conventional and organic *Coffea arabica* crops. Data were collected over five years in commercial *C. arabica* crops on seven 100 ha central pivots. The cost of control in organic crops was 16.98% higher than conventional. The decreasing order of control cost was manual > drone > airplane > tractor application. Coffee plants were tolerant to low densities (up to 15% mined leaves) of the pest that caused losses of up to 6.56%. At high pest densities (54.20% mined leaves), losses were high (85.62%). In organic and conventional crops and with the use of different insecticide application technologies, EIL and ET were similar. The EIL and ET were 14% and 11% of mined leaves, respectively. Therefore, these indices can be incorporated in integrated pest management programs in *C. arabica* crops. The indices determined as a function of insecticide application technology in organic and conventional coffee are important as they serve producers with different technological levels. Additionally, EILs and ETs can contribute to more sustainable production, as control methods will only be employed when the pest density reaches these indices.

## 1. Introduction

Coffee (*Coffea* spp.) is one of the most consumed drinks on the planet [1]. World coffee production in 2022 was 179.71 million 60 kg bags, comprising an area of 12.21 million hectares [2,3]. Coffee production involves 78 countries in the Americas (48.74% of production), Asia (32.39% of production), Africa (18.47% of production) and Oceania (0.40% of production). In addition to the economic aspects, coffee crops have social importance, as they create many direct and indirect jobs and are cultivated mainly by small farmers [2]. There are four species of coffee cultivated: *Coffea arabica* L., *Coffea canephora* L., *Coffea dewevrei* De Wild. & T. Durand and *Coffea liberica* Bull. *Coffea arabica* is responsible for around 60% of world coffee production [4].

The leaf miner *Leucoptera coffeella* (Guérin-Méneville) (Lepidoptera: Lyonetiidae) is one of the most important pests in coffee crops [5,6]. This pest is monophagous and attacks coffee crops in the Neotropical region and Africa [6,7,8,9]. During its life cycle, this insect progresses through the stages of egg, larva, pupa and adult. Its larvae undermine leaves, causing a reduction in plant photosynthesis and leaf senescence. It is reported that the attack by this pest can cause losses of up to 87% in the yield of coffee crops [6,9,10]. It has been verified that the densities of *L. coffeella* are higher in coffee crops grown in regions with higher air temperatures and a longer dry period, as occur in the Cerrado biome in Brazil [6,11]. Control of *L. coffeella* is carried out mainly with the application of insecticides. These applications can be carried out manually, or using a tractor or plane [12]. Recently, drones have also been used to apply pesticides to coffee crops [13,14].

The economic injury level (EIL) is the lowest density of the pest at which economic damages match the costs of control measures [15]. The economic threshold (ET) is the density of the pest at which control measures must be taken so that this population does not reach the EIL. EIL and ET are the main indices used in decision-making systems for integrated pest management programs. EIL is influenced by the value of production, cost and efficiency of pest control methods and the ability of these organisms to cause damage to plants [15,16]. EILs are inversely proportional to the value of production and directly proportional to the cost of pest control and the ability of these organisms to cause damage to plants [15,16,17]. The value of production is influenced by the crop yield and the value of the agricultural product. The cost of control is influenced by the cost of the products applied, the application technology and the number of applications [15,18,19]. Direct pests attack the marketed organs of plants, while indirect pests do not attack commercial organs. Therefore, *L. coffeella* is an indirect pest, as its larvae attack the leaves and not the beans, which are the part sold in coffee plantations [5,6]. The damage capacity of indirect pests is expressed by the type of curve that describes the yield of plants as a function of the density of these organisms [15,19]. 

In crops, pesticide application technology can influence pest control decision-making rates in two ways: due to their different costs and because they are used in different situations. In this context, pesticide applications using tractors generally have the lowest cost per hectare. Manual pesticide applications are the ones with the highest cost per hectare. Furthermore, when there is a need to control pests in large flat areas quickly, pesticide applications are carried out by plane [20,21]. Drones are a modern technology that can be used in both flat and mountainous areas [13,14].

The main production systems for *C. arabica* are conventional and organic. For a crop to be considered organic, it must be certified according to specific standards. Synthetic pesticides, fertilizers with high solubility (so-called chemical fertilizers) and genetically modified organisms cannot be used on these crops. In organic crops, chemical pest control is carried out through the application of natural insecticides, especially the use of botanical insecticides. To control *L. coffeella* in organic coffee crops, botanical insecticides could be applied in sprays [5]. In conventional crops, chemical pest control is carried out mainly through the application of organosynthetic insecticides [9,22,23]. To control this pest in conventional coffee crops, organosynthetic insecticides have been applied via soil and spray [9]. These products cause mortality during ingestion and can also have sublethal effects. Sublethal effects include reduction of oviposition by *L. coffeella* on treated plants and adult emergence [9,23,24,25].

The conventional agricultural production system, although widely adopted, raises several environmental and health concerns. The extensive use of chemical inputs such as synthetic fertilizers, pesticides and herbicides can lead to soil, water and even food contamination. In contrast, the organic production system, by adopting more sustainable methods, aims to promote soil health, biodiversity and the quality of water and food [26]. Measuring the EIL and ET indices for control decision-making *L. coffeella* in organic and conventional coffee plantations is crucial for more efficient and sustainable management of this pest. The use of these indices allows pest control to be carried out in the appropriate places and times. Furthermore, the use of these indices reduces production costs and the environmental impact of unnecessary use of control methods [22,27]. Despite the importance of coffee crops and *L. coffeella*, to date, no EILs nor ETs have been determined for this insect. The indices currently used in decision-making to control *L. coffeella* in coffee crops (20% to 30% of mined leaves) were suggested by researchers empirically [28]. Thus, the objective of this work was to determine EILs and ETs for *L. coffeella* as a function of insecticide application technology for conventional and organic *C. arabica* crops. 

## 2. Material and Methods

### 2.1. Design

The data used in the work were collected over five years (2015/2016 to 2020/2021) in seven central pivot irrigation systems with commercial *C. arabica* crops. These crops were located in the Cerrado biome in Barreiras, Bahia, Brazil (45°30′29.44″ W, 12°18′16.04″ S). Each pivot had 100 ha and the plants were of the red catuaí variety. The spacing used was 3 × 1 m and irrigation was carried out using a central pivot according to the plants’ water needs. Fertilization was carried out according to the nutrient content of the soil [29]. Each experimental unit consisted of a central pivot of 100 hectares containing 333,333 coffee plants. Thus, in the five years this study was conducted, 11.67 million coffee plants were used. The work was carried out in three stages. First, the pest control costs were determined. In the second, the crop yield curve was determined depending on the intensity of the pest attack. In the third stage, the economic injury levels (EIL) and economic thresholds (ET) were determined.

### 2.2. Pest Control Costs

In order to determine the cost of control for *L. coffeella*, a survey was carried out with the producers and technicians responsible for managing the coffee crops to find out the products (insecticides and adjuvants) used for *L. coffeella* control. The insecticides selected were those registered and most used to control *L. coffeella* in coffee crops [30]. Research was carried out on the number of annual insecticide applications to control *L. coffeella* in coffee crops in the Cerrado biome. This was carried out because this is the region in which *L. coffeella* has caused the greatest problems [11]. 

Research was carried out with companies that provide pesticide application services, technicians and coffee producers on the costs of applying insecticide by tractor, manual, plane and drone. All costs were taken into account. Research into the costs of applications by tractor, manual and airplane were carried out due to these being the main technologies used in pesticide applications on coffee crops [31,32]. The research into the costs of applying insecticides by drones was carried out due to this being a modern and promising technology for applying pesticides [13]. In conventional crops, applications of insecticides via soil can be carried out manually or with a tractor, with only one application being made. Foliar sprays are carried out manually, by tractor, plane and drone, with five applications being made. In organic crops, products are applied only by foliar spraying manually, and by tractor, airplane and drone. Six applications are made annually. 

Insecticide prices were researched in stores selling agricultural products in the Cerrado biome. The annual pest control costs were determined using Formula (1) [20,33]:Cij = (CIi + CAj) × Ni(1)
where C = annual control cost (USD ha^−1^), i = coffee cultivation system (1 = conventional and 2 = organic), j = coffee cultivation technology application (1 = manual, 2 = tractor, 3 = airplane and 4 = drone) of insecticide application, CI = average cost of insecticides (USD ha^−1^) used in an application, CA = cost of using each application technology and N = average annual number of insecticide applications.

### 2.3. Determination of the Crop Yield Curve Depending on the Intensity of Pest Attack

In the central pivots, the pest attack intensity was monitored in 100 plants, distributed in a regular grid. In each plant, four leaves belonging to the fourth-most apical pair of branches in the middle third of the plant canopy were evaluated. The presence of active mine (mine with live larvae) was assessed on each leaf (Figure 1). This was carried out because this is the methodology used to evaluate the intensity of *L. coffeella* attack on coffee crops [11,31].

The average monthly pest attack intensity was determined using Formula (2) [11]:AIpam = (100 × NLpam) ÷ 400(2)
where AI = attack intensity (%) of the pest, p = central pivot (1 to 7), a = year (1 to 5), m = month (1 to 12) and NL = number of leaves with active mines of *L. coffeella*. Subsequently, the average annual intensity of pest attack in each central pivot was calculated.

Mechanized harvesting was carried out annually on the coffee plantations in each central pivot. The harvested grains were processed and dried until they reached 11% moisture. The annual productivity was calculated in each central pivot (bags.ha^−1^). Productivity data (bags.ha^−1^) were transformed into yield (%) using Formula (3). This was performed due to requiring the yield value (%) of crops for the EIL calculation formula [15,34].
Ypa = (100 × Pdpa) ÷ MPd(3)
where Y = coffee crop yield (%), p = central pivot (1 to 7), a = year (1 to 5), Pd = crop productivity of coffee (bags.ha^−1^) and MPd = maximum coffee productivity during the period of work (82 bags·ha^−1^).

Data analyses were performed using R-4.1.3 and RStudio software 2023.12.0+369 [35]. Yield data (%) as a function of the average annual intensity of *L. coffeella* attack (percentage of mined leaves) were subjected to regression analysis at *p* < 0.05. The significant model (*p* < 0.05) was selected, based on which had the highest coefficient of determination (R2) and which had biological significance to describe the studied phenomenon [36,37]. The total degrees of freedom of this analysis were 34; the number of degrees of freedom of the model was 1 and the number of degrees of freedom of the error was 33. The confidence interval of the selected model at 95% probability was also determined.

### 2.4. Determination of Economic Injury Levels and Economic Thresholds

Economic injury levels and economic thresholds for *L. coffeella* were determined, depending on the insecticide application technology for organic and conventional *C. arabica* crops. The application technologies studied were manual, tractor, airplane and drone. The study of these first three technologies was due to them being those used by farmers in the application of insecticides on coffee crops. The study of applications using drones was due to this technology being promising for carrying out pesticide applications on coffee crops [13,14]. The production systems studied were conventional and organic. This was due to them being the two main coffee production systems worldwide [29].

Using the equation of the yield curve of coffee crops as a function of the intensity of attack by *L. coffeella*, Equation (4) was generated to calculate the EIL [15]:EILij = (Lij ÷ 0.029) × 0.5(4)
where EIL = economic injury level, i = coffee cultivation system (1 = conventional and 2 = organic), j = insecticide application technology (1 = manual, 2 = tractor, 3 = plane and 4 = drone) and L = losses caused by the pest (%) when its density is equal to the EIL.

Lij values were determined using Formula (5) [33]:Lij = (100 × Cij) ÷ (k × V0)(5)
where L = losses caused by the pest (%) when its density is equal to the EIL, i = cultivation system (1 = conventional and 2 = organic), j = insecticide application technology (1 = manual, 2 = tractor, 3 = airplane and 4 = drone), C = annual control cost (USD·ha^−1^), k = pest control efficiency and V0 = value average production of coffee crops without pest attack (USD·ha^−1^). 

The value k = 0.8 was used due to 80% being the control efficiency required for the registration of insecticides in Brazil [38]. The V0 value was determined using Formula (6) [15]:V0 = APd × CP(6)
where V0 = production value of coffee crops without pest attack (USD·ha^−1^), APd = average productivity of coffee crops (40 bags·ha^−1^) and CP = price of coffee (USD 187.05·bag^−1^).

The CP price value was the average price of coffee received by farmers from 2019 to 2023 [39]. These prices were corrected for October 2023 according to the inflation that occurred in Brazil during this period using the IPCA index [40].

ET were determined using Formula (7), according to [15,41]:ETti = EIL ti × 0.75(7)
where ET = economic threshold for the pest, t = cultivation system (1 = conventional and 2 = organic), i = insecticide application technology (1 = manual, 2 = tractor, 3 = airplane and 4 = drone) and EIL = economic injury level. In Formula (7), the coefficient of 0.75 was used due to the value of 25% (1.00 − 0.25 = 0.75) being the maximum error allowed in the determinations of the components in the decision-making systems of integrated pest management programs [42,43]. This allows decision-making related to *L. coffeella* control to be performed early, so that natural methods and applied products control the pest before it causes economic damage. In this context, Pereira et al. [6] report that natural agents (natural enemies, climatic elements, plant resistance and physiological disturbances during the life cycle) are responsible for mortalities of 89% to 94.3% of *L. coffeella* in commercial *C. arabica* crops.

The percentage values of the confidence intervals of the yield curve of coffee crops as a function of the attack intensity of *L. coffeella* were used to determine the confidence intervals of the EIL and ET (Figure 2A). When there was an intersection between these confidence intervals, it was considered that the compared decision-making indices did not present a significant difference (*p* < 0.05).

In the economic injury level and economic threshold recommendations, coffee crops were divided into three groups. In the first group were crops with flat topography (up to 12% slope), where the adoption of mechanization is possible. In the second group were crops in mountainous regions, where the adoption of mechanization is not possible. In the third group were the crops of family farmers with low purchasing power. This division was carried out due to these groups of crops being in situations that use different insecticide application technologies [12,20,21].

## 3. Results

### 3.1. Pest Control Costs

#### 3.1.1. Costs in Conventional Crops

To control *L. coffeella* in conventional *C. arabica* crops, insecticides are applied to the soil and in foliar sprays. On average, one application of insecticides is made to the soil and five foliar sprays. The active ingredients used the most in soil applications are imidacloprid and thiamethoxam. Insecticide applications to the soil can be carried out manually or using a tractor. The total cost of manual application of insecticide to the soil is USD 154.14 per hectare, with 19.95% of this cost due to manual application and 80.05% due to insecticide expenditure. The total cost of applying insecticide to the soil using a tractor is USD 140.85, with 12.37% of this cost due to the use of the tractor and 87.63% due to the expense of the insecticide (Table 1).

The insecticides used the most in foliar sprays to control *L. coffeella* in *C. arabica* crops are Cartap, Chlorantraniliprole, flupyradifurone and mixtures of abamectin and chlorantraniliprole as well as lufenuron and profenofos. The total cost of manual spraying is USD 314.15; 32.80% of this cost is due to manual spraying and 67.20% is due to insecticide costs. The total cost of spraying carried out using a tractor is USD 247.70; 25.97% of this value is due to the use of the tractor and 74.03% due to insecticide costs. The total cost of applications carried out with an airplane is USD 257.95; 27.36% of this value is due to the use of the airplane and 72.64% due to the insecticides applied. The total cost of applications carried out with a drone is USD 298.85; 31.60% of this value is due to the use of the drone and 68.40% due to insecticides (Table 2).

#### 3.1.2. Costs in Organic Crops

The two natural products registered in Brazil for the control of *L. coffeella* in coffee crops are the botanical insecticides *Sophora flavescens* and azadaractin. These insecticides are applied in approximately six sprays annually. The total cost of control using natural manual applications is USD 536.76. Of this cost, 25.53% is due to manual spraying activities and 74.47% is due to expenses with natural insecticides. The total control cost using a tractor is USD 457.02. Of this cost, 18.58% is due to the use of the tractor and 81.42% due to expenses with natural insecticides. The total control cost using an airplane is USD 469.32. Of this cost, 19.89% is due to the use of the plane and 80.11% is due to expenses with natural insecticides. The total cost of control using a drone is USD 518.40. Of this cost, 24.21% is due to the use of the drone and 75.79% is due to expenses with natural insecticides (Table 3).

### 3.2. Crop Yield as a Function of Pest Attack Intensity

The regression curve of the yield of *C. arabica* crops as a function of the percentage of leaves mined by *L. coffeella* was significant (F 1, 33 = 41.60 and *p* < 0.0001) and it presented a coefficient of determination of 59%. This model was quadratic, with concavity facing downwards. At low intensities of attack by *L. coffeella*, yield losses in coffee crops were low. This can be exemplified by the fact that, when the percentages of leaves mined by *L. coffeella* were 5%, 10% and 15%, losses in crop yield were 0.73%, 2.91% and 6.56%, respectively. At the maximum intensity of the pest attack (54.20% of mined leaves), the losses caused by the pest in the yield of coffee crops were high, that is, 85.62% (Figure 2A).

Figure 2B represents the residual analysis of the data from the regression model determined in this work. The average error of the regression model estimates was 8.72%, with a maximum of 27.03%. In the attack intensity range of up to 15% of leaves mined by *L. coffeella*, the average error of the model estimates was 8.37%, with a maximum error of 15.65%. In the range of attack intensity from 15.01% to 30% of leaves mined by *L. coffeella*, the average error of the model estimates was 8.77%, with a maximum error of 27.09%. In the attack intensity range of more than 30% of leaves mined by *L. coffeella*, the average error of the model estimates was 5.53%, with a maximum error of 20.96%.

### 3.3. Economic Injury Levels and Economic Thresholds

For conventional *C. arabica* crops, the EILs were 14.65%, 13.34%, 13.52% and 14.19% of leaves mined by *L. coffeella* for manual, tractor, airplane and drone applications, respectively. The average ET values were 10.99%, 10.01%, 10.14% and 10.65% of leaves mined by *L. coffeella* for manual, tractor, airplane and drone applications, respectively (Table 4).

For organic *C. arabica* crops, the EILs were 15.69%, 14.48%, 14.67% and 15.42% of leaves mined by *L. coffeella* for manual, tractor, airplane and drone applications, respectively. The average ETs were 11.77%, 10.86%, 11.00% and 11.56% of leaves mined by *L. coffeella* for manual, tractor, airplane and drone applications, respectively (Table 4).

Examining the confidence intervals at 95% probability of EIL and ET, it can be seen that these indices did not differ significantly (*p* < 0.05) with the insecticide application technology and coffee cultivation systems. The average EIL value was around 14% of leaves mined by *L. coffeella*. The average ET value was 11% of leaves mined by *L. coffeella* (Table 4).

## 4. Discussion

The fact that the four insecticide application technologies have different costs has implications for integrated *L. coffeella* management programs. The use of a tractor was the lowest-cost application technology due to the lower cost of using this machine (USD 34.76·h^−1)^ and its speed, as it only takes 30 min to spray 1 hectare of coffee with a tractor. Application by plane was the second application technology in terms of cost due to its speed (26 s·ha^−1^), despite its high cost per hour of use (USD 2689.95·h^−1^). Pesticide applications by plane are used in large areas with flat topography [20,21].

The cost of using a drone was the third application technology in terms of cost due to the high cost per hour of using this machine (USD 110.43·h^−1^), although this application is carried out quickly (15 min·ha^−1^). Due to the high cost of using a drone, it should only be used in special situations such as coffee plantations in mountainous regions [13], where it can carry out spraying to control *L. coffeella* quickly and with less cost than manual applications. Manual application was the application technology with the highest cost, due to the longer time spent on these applications (8 h·ha^−1^). This implies that small coffee farmers who apply pesticides manually, especially in mountainous regions, have the highest cost of controlling *L. coffeella*.

The component that most influenced the cost of controlling *L. coffeella* was the price of insecticides. In conventional coffee crops, the cost of insecticides corresponds to 71.45% of the cost of controlling the pest. In conventional crops, insecticides were applied to the soil (29.69% of this cost) and in spraying (70.31% of this cost). This happened since there was only one application to the soil but there were five annual sprays. In organic production systems, the cost of natural insecticides represented 87.50% of the cost of controlling *L. coffeella*, and, in these systems, insecticides are applied by spraying.

Crop yield curves depending on the intensity of pest attack indicate the degree of susceptibility of plants to these herbivores. There are three types of curves that describe crop yields as dependent on the intensity of pest attack. The first type of curve has an upward concavity, and it represents plants that are highly susceptible to insect pests. The second type is a straight line with a negative angular coefficient, and it represents plants susceptible to insect pests. The third type of curve has a downward concavity, and it represents plants tolerant to insect herbivory [36,44]. Therefore, coffee plants showed tolerance to *L. coffeella* since the yield curve of these plants as a function of the attack intensity of the pest had a downward concavity. *L. coffeella* injuries are due to the formation of mines in the leaves, which reduces photosynthesis and increases leaf senescence [45]. In this work, it was found that, when the percentage of leaves mined by *L. coffeella* was up to 15%, there was a reduction in the yield of coffee plants of only 6.56% Plant tolerance to insect attack is related to physiological mechanisms in plants. These mechanisms allow attacks by these herbivores to affect plant yield less [46].

Based on EIL, ET and control costs, decision-making systems for *L. coffeella* in coffee crops can be divided into three groups. The first group contains decision-making systems for crops with flatter topography and where the adoption of mechanization is possible. The second group has the decision-making systems for coffee plantations located in mountainous regions where the adoption of mechanization is not possible. The third group has the decision-making systems for the crops of family farmers with low purchasing power.

In crops with flatter topography, control of *L. coffeella* must be carried out with the application of insecticides by tractor. This minimizes the cost of controlling the pest, as this application technology has a lower cost. In this first group of crops, if there are some fields missing where it was not possible to control the pest, insecticide applications must be carried out using an airplane. This must be performed because the use of airplanes is quick to execute and this application technology had the second-lowest cost. In this group of crops, *L. coffeella* must be controlled when the density of this pest reaches 10.01% or 11.86% of mined leaves in conventional or organic crops, respectively, since these are the ETs in these production systems with the application of insecticides using tractors.

In coffee plantations in mountainous regions, control of *L. coffeella* must be carried out with the application of insecticides using a drone. This must be performed because this application technology can be used on crops in mountainous regions [13]. Furthermore, the use of a drone was less expensive than the manual application of insecticides. In this group of crops, *L. coffeella* must be controlled when the density of this pest reaches 10.65% or 11.56% of mined leaves in conventional or organic crops, respectively, since these are the ETs in these production systems with insecticide application by drone.

In the coffee plantations of family farmers with low purchasing power, control of *L. coffeella* is carried out manually using family labor [47]. In this group of crops, *L. coffeella* must be controlled when the density of this pest reaches 10.99% or 11.77% of mined leaves in conventional or organic crops, respectively. This must be performed because these are the ETs for these production systems with manual application of insecticides.

The economic injury levels (EIL) and economic thresholds (ET) were similar between production systems (conventional and organic) and insecticide application technologies (manual, tractor, airplane and drone), which has implications for the decision-making process for control of *L. coffeella*. In this context, the average EIL and ET values were 14% and 11% of leaves with active mines (with live larvae) of *L. coffeella*. Therefore, in both organic and conventional coffee crops, when the *L. coffeella* population reaches 11% of leaves with active mines, this pest must be controlled. If this decision is not adopted, this pest will cause economic damage. On the other hand, when the density of *L. coffeella* is less than 11% of leaves with active mines, insecticides (natural or organosynthetic) should not be applied to control this pest. In this situation, if insecticides are applied, production costs will increase and there will be an unnecessary environmental impact [15,22,48].

The EIL and ET determined in this work are important for coffee farming. These decision-making indices are the first determined for this pest using appropriate scientific criteria. These indices are robust and representative of the phenomena studied because they were determined in commercial coffee plantations over a long period (five years). Furthermore, the size of the area used to conduct the work (700 hectares) is representative for the determinations made and it is located in the region where *L. coffeella* causes the greatest damage, which is the cerrado biome [11]. Currently, the rates used in decision-making to control *L. coffeella* in coffee crops are 20% to 30% of mined leaves, as suggested by researchers empirically [28]. The indices defined in this work for organic and conventional production were much lower than those determined empirically. Making a comparison between these indices, coffee producers were carrying out *L. coffeella* control late, which resulted in productivity losses. In this way, using the appropriate control indices determined in this study, producers will make a safe and efficient control decision, reducing losses due to *L. coffeella* attack. Furthermore, with these indices, there is no excessive use of chemicals, ensuring sustainability in production and the preservation of the environment, including fauna, soil and water [18].

## 5. Conclusions

The economic injury levels (EILs) and economic thresholds (ETs) determined in this work for Leucoptera coffeella should be incorporated into integrated management programs in coffee crops. This must be carried out because these decision-making indices are robust, are representative of reality and have been determined using appropriate scientific methodology. In organic and conventional crops and with the use of different insecticide application technologies, EIL and ET were similar. The EIL and ET were 14% and 11% of mined leaves, respectively.

## Figures and Tables

**Figure 1 plants-13-00585-f001:**
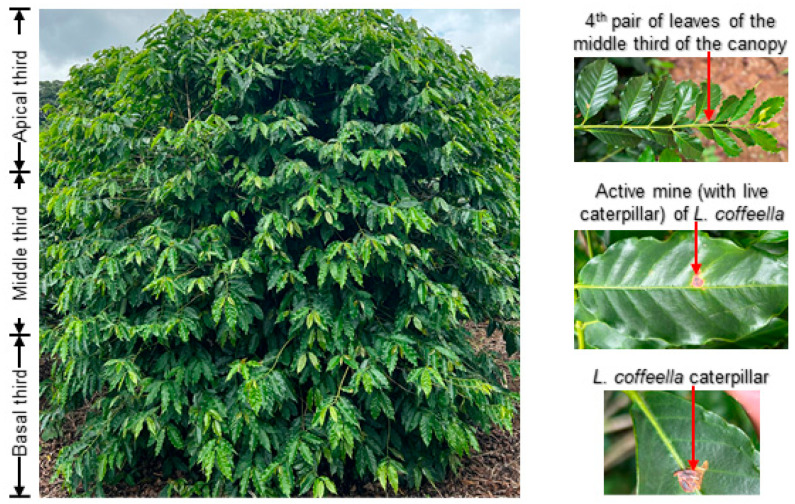
Thirds of the canopy of a *Coffea arabica* plant, pair of leaves used to evaluate attack intensity, active mine (with live caterpillar) and *Leucoptera coffeella* caterpillar.

**Figure 2 plants-13-00585-f002:**
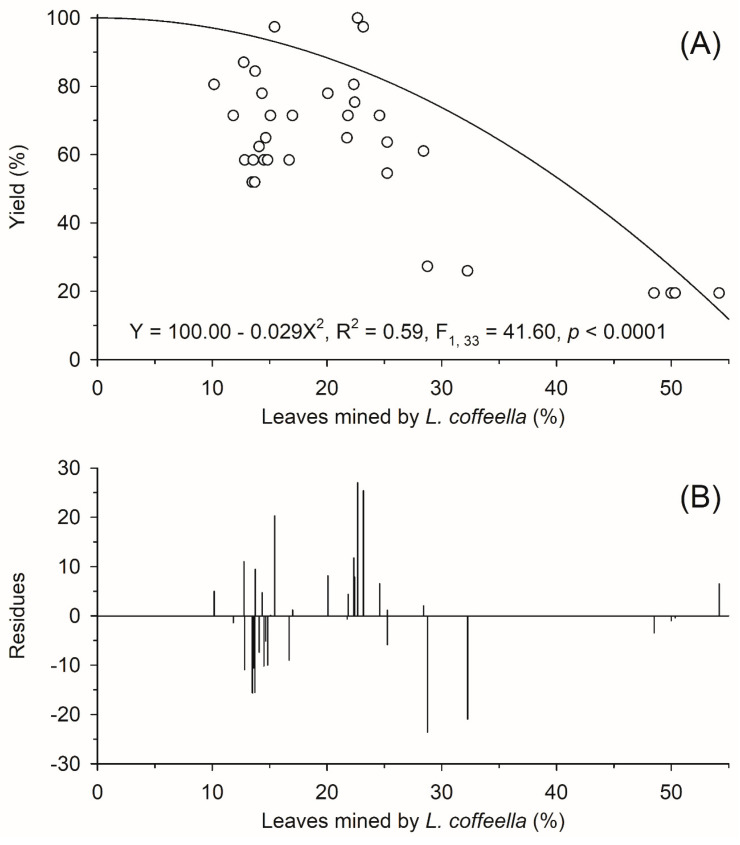
(**A**) Regression curve of the yield of *Coffea arabica* crops as a function of the attack intensity of Leucoptera coffeella and (**B**) residue analysis of the regression model data in (**A**); each circle represents a central pivot of 100 ha containing 333,333 coffee plants and the vertical straight segments represent the model’s confidence interval at 95% probability.

**Table 1 plants-13-00585-t001:** Costs of applying insecticide to the soil to control *Leucoptera coffeella* in conventional coffee crops.

Items	Unit	Unit Cost (USD)	Qty	Total Cost (USD ha^−1^)
Insecticides				
Imidacloprid 700 WG	kg	143.15	1.13	161.76
Thiamethoxam 250 WG	kg	50.10	1.70	85.17
(1) Average cost of insecticides applied to the soil				123.47
Manual application				
(2) Application cost				30.67
(3) Total cost of a manual application = (1) + (2)				154.14
Tractor application				
(4) Application cost				17.38
(5) Total cost of an application with a tractor = (1) + (4)				140.85

In the header: Qty = quantity of the item used per hectare in an application.

**Table 2 plants-13-00585-t002:** Costs of insecticide spraying to control *Leucoptera coffeella* in conventional coffee crops.

Items	Unit	Unit Cost (USD)	Qty	Total Cost (USD ha^−1^)
Insecticides				
Abamectin 18 SC + Chlorantraniliprole 45 SC	L	81.80	0.50	40.90
Chlorantraniliprole 350 WG	kg	153.37	0.09	13.80
Cartap 500 PS	kg	38.83	0.90	34.95
Flupyradifurone 200 SL	L	60.53	0.75	45.40
Lufenuron 50 EC + Profenofos 500 EC	L	36.81	0.70	25.77
(1) Average cost of insecticides				32.16
Manual application				
(2) Application cost				30.67
(3) Cost of an application = (1) + (2)				62.83
(4) Total cost = (3) × 5 sprays				314.15
Tractor application				
(5) Application cost				17.38
(6) Cost of an application = (1) + (5)				49.54
(7) Total cost = (6) × 5 sprays				247.70
Airplane applications				
(8) Application cost				19.43
(9) Cost of an application = (1) + (8)				51.59
(10) Total cost = (9) × 5 sprays				257.95
Drone applications				
(11) Application cost				27.61
(12) Cost of an application = (1) + (11)				59.77
(13) Total cost = (12) × 5 sprays				298.85

In the header: Qty = quantity of the item used per hectare in an application.

**Table 3 plants-13-00585-t003:** Costs of controlling *Leucoptera coffeella* in organic coffee crops.

Items	Unit	Unit Cost (USD)	Qty	Total Cost (USD ha^−1^)
Insecticides				
* Sophora flavescens* 190.5 SL	L	32.71	1.10	35.98
Azadaractin 12 EC	L	30.22	2.70	81.59
(1) Average cost of insecticides				58.79
Manual application				
(2) Application cost				30.67
(3) Cost of an application = (1) + (2)				89.46
(4) Total cost = (3) × 6 sprays				536.76
Tractor application				
(5) Application cost				17.38
(6) Cost of an application = (1) + (5)				76.17
(7) Total cost = (3) × 6 sprays				457.02
Airplane applications				
(8) Application cost				19.43
(9) Cost of an application = (1) + (8)				78.22
(10) Total cost = (6) × 6 sprays				469.32
Drone applications				
(11) Application cost				27.61
(12) Cost of an application = (1) + (11)				86.40
(13) Total cost = (12) × 6 sprays				518.40

In the header: Qty = quantity of the item used per hectare in an application.

**Table 4 plants-13-00585-t004:** Control costs, economic injury levels (EILs) and economic thresholds (ETs) for *Leucoptera coffeella* in conventional and organic coffee crops using different insecticide application technologies.

Insecticide Application Technology	Pest Control Costs (USD ha^−1^ Year^−1^) *			Mined Leaves (%) ^§^	
	Soil Application	Sprays	Total	EIL	ET
	Conventional crops				
Manual (soil application and sprays)	752.00	1536.41	2288.41	14.65 (13.76–15.54)	10.99 (10.32–11.66)
Tractor (soil application and sprays)	687.00	1211.41	1898.41	13.34 (12.51–14.17)	10.01 (9.38–14.17)
Tractor (soil application) and airplane (sprays)	687.00	1261.41	1948.41	13.52 (12.68–14.36)	10.14 (9.51–14.36)
Tractor (soil application) and drone (sprays)	687.00	1461.41	2148.41	14.19 (13.32–15.06)	10.65 (9.99–15.06)
	Organic crops				
Manual (sprays)	-	2625.06	2625.06	15.69 (14.75–16.63)	11.77 (11.06–16.63)
Tractor (sprays)	-	2235.06	2235.06	14.48 (13.59–15.37)	10.86 (10.19–15.37)
Airplane (sprays)	-	2295.06	2295.06	14.67 (13.77–15.57)	11.00 (10.33–15.57)
Drone (sprays)	-	2535.06	2535.06	15.42 (14.49–16.35)	11.56 (10.87–16.35)

* These values are from Table 1, Table 2 and Table 3. ^§^ These determinations were carried out using the curve in Figure 2, the EIL determination formula and the production value of USD 7482.13 per hectare. The numbers before the parentheses are the average EIL and ET values. The numbers in parentheses are the 95% probability confidence intervals for these indices.

## Data Availability

Data are contained within the article.

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
