# Peer review of "Economic Injury Levels and Economic Thresholds for *Leucoptera coffeella* as a Function of Insecticide Application Technology in Organic and Conventional Coffee (*Coffea arabica*), Farms"

_plants, 2024, doi:10.3390/plants13050585_

Round 1
Reviewer 1 Report
Comments and Suggestions for Authors
In general, this manuscript corresponds to the theme of the journal 'plant-insect interactions. This study presents the use of two indices, EIL and ET, for pest management decisions on L. coffeella, an important pest of coffee, using 4 types of techniques (manual, tractor, airplane, and drone) on organic and conventional coffee farms.
The data from the study are also sufficient and acceptable. However, there are a few things that the author needs to check: i) the number of study samples (group and overall, especially in relation to the regression analysis), ii) factors discussed in the discussion section but not presented in the methodology, and iii) the use of the decision-making indices EIL and ET for different technologies/techniques is also not particularly emphasized in the introduction/abstract.
Comments on the Quality of English Language
It is also recommended to 'proofread' the manuscript as it contains some sentences that need to be rephrased in some sections.
Author Response
Reviewer 1
The data from the study are also sufficient and acceptable. However, there are a few things that the author needs to check:
- the number of study samples (group and overall, especially in relation to the regression analysis),
As described in Material and Methods (lines 104 - 106), 35 experimental units were used in this part of the work. Each experimental unit consisted of a central pivot of 100 hectares containing 333333 coffee plants. In the five years this study was conducted, 11.67 million coffee plants were used. Therefore, this must have been one of the pest control decision-making index studies that used one of the largest areas and was conducted over one of the longest periods of time. The factors under study were the intensity of the pest attack (percentage of leaves with active mines) and the yield of coffee plants in each pivot. The total number of degrees of freedom used in the statistical analyzes in this part of the work was 34 (35-1), the number of model degrees of freedom was 1 and the number of error degrees of freedom was 33. This information is presented in Figure 2 (lines 292-297). Therefore, the model determined to study mild economic injuries and economic thresholds are robust. In Figure 2 (lines 292-297) and in Material and Methods (lines 167 - 170) information was inserted to make this part of the work clearer.
- factors discussed in the discussion section but not presented in the methodology, and
Changes were made to the Material and Methods (lines 222 - 229) to meet this request.
- the use of the decision-making indices EIL and ET for different technologies/techniques is also not particularly emphasized in the introduction/abstract
Changes were made to the Introduction (lines 69 - 76) to meet this request.
Abstract
- Suggestion - Briefly explain why your result is important for the pest control programme for L. coffeella on organic versus conventional coffee farms.
Thanks for the suggestion. Done.
- In addition to using the EIL and ET indices, and based on your results, you may as well highlight the advantages and disadvantages of using manual, tractors, drone and airplanes technologies/techniques for insecticide application.
Thanks for the suggestion. Done.
Title
- Suggestion; Economic injury levels and economic thresholds for Leucoptera coffeella as a function of insecticide application technology in 3 organic and conventional coffee, Coffea arabica farms.
Thanks for the suggestion.
Introduction
- L64 – L66.Suggestion for improvement: Direct pests attack the marketed organs of plants, while indirect pests do not attack commercial organs.
Thanks for the suggestion.
- L71 – L72. ‘To control L. coffeella in organic coffee crops, botanical insecticides can be applied in sprays. To control this pest in conventional coffee crops, organosynthetic insecticides are applied via soil and spray’. The description in this section should be in the 'past tense', as this is a common practice or is already implemented or commonly applied by farmers.
Thanks. Done.
- Please add a description of the botanical and organosynthetic effects of insecticides on L. coffeella? Can the use of botanical insecticides reduce the pest infestation of L. coffeela in organic coffee plantations? Can the use of organosynthetic insecticides also demonstrably reduce the pest infestation of L. coffeela in conventional coffee plantations?
A sentence and literature were added about the effect of these products on l. coffeella.
- The two types of insecticides used on organic and conventional farms also have different classes, modes of action and modes of entry. Is it suitable for the control of L. coffeella pests in coffee plantations? How can you be sure that the insecticide is suitable to control the pest infestation, which in turn also has an equal impact on the EIL and ET indices calculated in the methodology section?
The two insecticides used to determine economic injury levels for L. coffeella in organic coffee crops are the only natural insecticides registered in Brazil for the control of this pest. In the process of registering insecticides for pest control in crops in Brazil, research is required to be carried out in three different locations. For the product to be registered, it is required that in these experiments they present a minimum efficiency of 80% in controlling the pest. In this sense, in calculating the economic injury levels determined in this work, an efficiency of 80% (k = 0.8) was used (lines 195 - 196).
- How does this section demonstrate the importance of using EIL and ET indices using 4 different application techniques (manual/tractor/drone/airplane)?
This part is included in the introduction section.
Materials and methods
- The section on methodology, especially experimental design, is not clear and needs to be improved. Several factors discussed in the discussion section are not included in the methodology section (see further comments on the discussion section).
Modifications were made to the text of the Material and Methods to meet the reviewer's suggestion.
- Section 2.1. – Please add Location of Studies.
Add. Barreiras, Bahia, Brazil (45°30′29.44″ W, 12°18′16.04″ S).
Section 2.1 - 2.2
- L82. Experimental characteristics – ‘design’
Done.
- Please briefly describe the pest control system for coffee pests, L. coffeella used in organic and conventional coffee plantations. Do you mean that no chemical insecticides are used at all for pest control in organic coffee farms? Has the area been recognized by the Department of Agriculture or relevant authorities as an organic certified coffee growing area, e.g. as part of ‘Good Agricultural Practice’ certified farm? How is the coffee growing area categorized as “organic” or “conventional” in your study?
Modifications were made to the text of the Introduction (lines 77 - 84) to meet the reviewer's suggestion.
Section 2.2 - 2.3
- L92 – L93. How are insecticides selected for pest control on conventional and organic coffee farms? Are they based on chemical classes, mode of action and mode of entry? How are these insecticides considered suitable to reduce the population of L. coffeella?
Research was carried out on insecticides used to control L. coffeella in conventional and organic coffee crops with coffee producers and professionals who provide consultancy to coffee producers. The insecticides selected were those registered and most used to control L. coffeella in coffee crops.
- Table 1. ‘Technologies’ – is ‘manual application of insecticides’ (e.g labour – human) considered a technology? Suggestion to change ‘technologies’ to ‘techniques’.
Describe the technologies directly in the text. Removed table 1.
L112 – L114. Is manual application associated with labour costs? Use of tractors – including hire, driver, fuel? Use of airplanes – including driver, fuel, rental, etc.? Use of drones – rent, electricity costs? How are these factors taken into account in the cost calculation of the indices?
Manual application is not associated only labour costs. Manual application consists of a human being with a sprayer on his back applying insecticides, widely used in family farming, when it is not possible to buy a tractor. All costs were taken into account.
- Suggestion for inclusion in the introductory section: Why do you think measuring EIL and ET indices on organic and conventional coffee farms is important for managing pest, L. coffeella infestations? How does this relate to the decision to use / type of chemical and botanical insecticides and as well using different insecticides application techniques? See L198 – L214.
Thanks for the sugestion. This part was included in the introduction.
- L198-L199. “ The insecticides most used in foliar sprays to control coffeella in C. arabica crops are Cartap, chlorantraniliprole and flupyradifurone and mixtures of abamectin + chlorantraniliprole and lufenuron + profenofos”.
Research was carried out on insecticides used to control L. coffeella in conventional and organic coffee crops with coffee producers and professionals who provide consultancy to coffee producers. The insecticides selected were those registered and most used to control L. coffeella in coffee crops.
- L213 – L214. “The two natural products registered in Brazil for the control of coffeella in coffee crops are the botanical insecticides Sophora flavescens and azadaractin”.
Research was carried out on insecticides used to control L. coffeella in conventional and organic coffee crops with coffee producers and professionals who provide consultancy to coffee producers. The insecticides selected were those registered and most used to control L. coffeella in coffee crops.
Section 3.2
L.230 – L231. The regression curve of the yield of C. arabica crops as a function of the percentage of leaves mined by L. coffeella was significant (F 1, 33 = 41.60 and p < 0.0001) and it presented a coefficient of determination of 59%.
- Please check and verify the p-value given in L230-L231, p < 0.0001. What is the highest p-value that indicates a significant difference? p < 0.05 / p < 0.01 / p < 0.001. Don’t you think that the p-value in your study is too high?
This p value = 2.58 x 10-7. Therefore, this p value was less than 0.0001 as reported in Figure 2. This fact demonstrates that the pest control decision-making indices determined in this work are robust.
- F 1,33, What is the size of your group (2-1 = 1) and the total number of samples (33?? Please check)? How did you get this value – 33? How is the intensity of pest infestation measured for 100 plants – see L118.
Thirty-five experimental units were used. Each experimental unit consisted of a central pivot of 100 hectares containing 333333 coffee plants. The total number of degrees of freedom used in the statistical analyzes in this part of the work was 34 (35-1), the number of degrees of freedom for the model was 1 (the model has a coefficient), and the number of degrees of freedom for the error was 33. In each of the five years, monthly assessments of the intensity of the pest attack were carried out. Each month, 100 plants were evaluated. Four leaves were evaluated on each plant. It was assessed whether or not the leaf had an active mine (mine with live larvae). These parts of the Material and Methods have been changed to make these parts of the methodology clearer.
- Please also check and verify the p value given in Figure 2, p < 0.0001 (L246)
This p value = 2.58 x 10-7. Therefore, this p value was less than 0.0001 as reported in Figure 2. This fact demonstrates that the pest control decision-making indices determined in this work are robust.
Discussion
- It is recommended that the discussion of the study is divided into several sections that correspond to the order of the study results.
- The decision-making indices (EIL and ET), which discuss the overall result, can be placed in the last paragraph so that the reader understand what the author is trying to convey in an organized way.
Ok.
- L341 – L347. The description of the study results that can demonstrate the significance of the study should be briefly emphasized in the Introduction/Methods section.
Ok.
- The factors that support the discussion of the study's findings and are discussed in the Discussion section should be included in the Methodology section as follows.
- L320-L321. The topographical nature of the study area (flat, mountainous, etc.) should be included in the methodology section.
- L336 – L337. The description of the manual technique with family labour should be included in the methodology section.
- A description of the decision-making indices EIL/ET by type of technology/insecticide application technique is also suggested for emphasis.
Text was included in the Material and Methods to meet this reviewer's suggestion.
Reviewer 2 Report
Comments and Suggestions for Authors
Overall
The present manuscript investigates the damages from Leucoptera coffeella on Arabica coffee, the implementation costs of different control solutions, and estimates economic thresholds to guide coffee growers in their management of the pest. Overall, the topic is of high interest as it combines a scientific output with a practical output for coffee growers. Furthermore, the data collected for this research (5 years data on 7 farms and 100 coffee trees per farm) provides solid foundation for the conclusions. I would suggest several major revisions to this manuscript, to improve its scientific value and clarity before publication.
Overall major revisions
1. Adding a measure of uncertainty. All figures presented in the manuscript are presented as right on the spot. However, there is uncertainty associated with these values, normally indicated by standard error or standard deviation in the case of averages. Without this measure of uncertainty, there is no way to know how trustful the results are, and there is no possible comparison between values. For instance, how can we know if the EIL for conventional coffee and tractor application is right on 13.34% or actually between 13 and 14% or even between 10 and 16%? How can we know if the EIL for conventional coffee (13.34%) is really lower (= statistically lower) than the EIL for organic coffee (14.48%)?
2. Beware of significant numbers (this comment is also linked with uncertainty, see comment 1). Figures expressed throughout the manuscript are expressed with too many digits. I doubt that the authors can say with certainty that the EIL for conventional coffee and tractor application is right on 13.34%. I believe that 13.3% is already more plausible, and that 13% might actually be accurate enough with this experimental design and for practical recommendations.
3. Crop yield as a function of pest attack intensity. Looking at figure 2a, I do not see a quadratic function as the best fit for regression. Almost all of the data points are below the quadratic curve. I am convinced that a linear regression would be a better fit. Furthermore, figure 2b does not match with figure 2a. In figure 2b, the residues are presented are equally distributed between positive values and negative values, but in figure 2a, almost all the data points are below the quadratic curve and therefore almost all of the residues are negative. The distance between the curve and the data points reaches at least 40% in some case, far above the claim 27.03%.
This issue with the regression is all the more important that the results of this curve are then vastly extrapolated. For instance, the authors claim that there is less than 1% damage at 5% of pest intensity, and less than 3% at incidence of 10%, but there is no evidence of that. This claim is only based on the regression, not on data points (the lowest incidence recorded on farm is only a single point around 10% and with damages of around 20%). The authors also rely on the shape of the curve to claim that coffee is tolerant to L. coffeella. And this curve is of primary importance to calculate the indices EIL and ET.
4. Equations and coefficients used in section 2.4 on economic thresholds. I was not familiar with the indices EIL and ET before the review of this article, but a rapid research helped me understand the concepts. Based on my understanding, I do not believe that the equations 4 to 7 are correct to calculate EIL and ET. In the current version of the manuscript, I do not understand how the authors proceed with calculations, and where they get some of their coefficients.
Equation 4: by definition, EIL is related to both economic losses from pest damages (L) and costs for pest control (C). Both terms must appear in equation 4, but this is currently not the case. Equation 4 should look like EIL = min(infestation) with costs from pest (L) = cost from treatment (C). Where do coefficients 0.029 and 0.5 come from?
Equation 5: economic losses (L) by the pest are not related to control costs (C), they are related to losses in yield (Y). Equation 5 should look like losses = CP*V0*(1-Y). Why is V0 taken as 40 bags/ha? On figure 2, it looks like most farms with pest incidence < 20% are on average around 70% of maximum yield (82 bags/ha), hence V0 ~ 57 bags/ha.
Moreover, the cost of control is defined in section 2.2. If efficacy of the measures are only 80%, this must be translated at residual damage from pest. In that case, we could have something like economic losses after treatment = 20% * L.
Finally, ET is arbitrarily set at 75% of EIL, meaning that actions are to be taken “25% before the EIL threshold”. That’s ok, but the authors should explain this in the text, to help the reader understand. As it stands, it can appear to the reader that ET and EIL are calculated separately. While in fact, only EIL is calculated based on empirical evidence, and ET is simply set at 75% afterwards as a prevention measure.
Introduction
The introduction is clear, well written and backed with good references. There are a few minor comments to improve it:
- L.29: coffee is not the 2nd most traded commodity. Oil, iron, gold and many more are above coffee.
- L.36: FAOstat (= reference [2]) does not give figures about coffee growers/consumers.
- L.51 and definition of EIL: if I am not mistaken, EIL is defined as the lowest density of pest at which economic damages match the costs of control measures. Please check the definition.
- L.76: the previous indices based on expert knowledge and empirical evidences (20 to 30%) are key to this article. They must be compared to the values found by the authors and discussed later on in the article.
Material and Methods
- Please provide more information on the farm area: how were they selected, what is the climate there (especially since the authors mentioned climate as an important factor for damages in the introduction), what is the average productivity, and so on.
- L.83: I was not familiar with central pivots. I suggest to replace with “central pivot irrigation systems”, which is more easily understandable, at least the first time that you mention central pivots.
- 2.2 Pest control costs: the authors repeat several times “research was carried out” but do not explain how it was carried out. What was the method of data collection (interviews, online shops websites, existing guidelines, literature review, ...)? How were the sources of information selected, and how many replicates were used to confirm the information? What were the variables investigated (cost per product, amount of products to use, possible methods of applications,…)?
- Table 1 does not seem necessary. Instead, I suggest to describe the possible technologies directly in the text. This was done in l.185-188 and l.213.215. These sections would be better suited in the material and methods.
- 2.3 Determination of crop yield curve. The amount of work necessary to collect this dataset over 5 years must be huge and provides solid ground for this research.
- Equation 3. Yield is a measure expressed in kg or bags per unit area, not in percentage. If the authors want to express it in percentage, they must name it differently. Furthermore, where does MPd = 82 bags/ha come from?
Results
- The current presentation of the conventional and organic control costs is clear and can be kept as such. Nonetheless, I want to suggest an alternative and more concise representation of these costs. In my opinion, table 2 could aggregate the costs of insecticides for conventional treatments (top section of current table 2) and organic treatments (top section of current table 3). Table 3 could list the total cost of each treatment in a format like:
|
|
Avg cost of Insecticide (US/ha) |
Application method |
Cost of application (US/ha) |
# of application |
Total cost of application (US/ha) |
Total cost (US/ha) |
|
Conventional |
123.5 |
Manual |
30.7 |
5 |
153.5 |
277 |
|
conventional |
123.5 |
Tractor |
17.4 |
5 |
87 |
… |
|
organic |
32.2 |
Manual |
30.7 |
6 |
184.2 |
… |
- In addition, I suggest the authors to add a graphical representation of EIL and ET, at least for one of the treatment. This could be a line representing cost from treatment, a line representing the cost from pest damage. EIL will be at the intersection of these lines. This would help reader to understand the equations used in the manuscript and the meaning of these decision-support indices.
Discussion
- The current discussion section is very pragmatic-oriented, and discussed the use of different application methods depending on farmers’ context. It is sufficient as it is, if the main goal of the authors is to provide decision-support indices for farmers. On the other hand, from a scientific point of view, I would suggest the authors to discuss the results in the larger context of integrated pest management. For instance, it has been shown with coffee leaf rust disease, that damages in year 1 result in yield losses in year 1, but also year 2 and 3 (see Cerda, R., Avelino, J., Gary, C., Tixier, P., Lechevallier, E., & Allinne, C. (2017). Primary and Secondary Yield Losses Caused by Pests and Diseases: Assessment and Modeling in Coffee. PLoS ONE, 12(1), e0169133. doi:10.1371/journal.pone.0169133). The authors could also quickly discuss the environmental and sanitary impacts from the different treatment methods.
Conclusion
- Are the EIL and ET really different in organic and conventional management?
- If the goal is to come up with practical guidelines for farmers, shouldn’t the thresholds be simplified and rounded up or down?
Author Response
Reviewer 2
- Adding a measure of uncertainty. All figures presented in the manuscript are presented as right on the spot. However, there is uncertainty associated with these values, normally indicated by standard error or standard deviation in the case of averages. Without this measure of uncertainty, there is no way to know how trustful the results are, and there is no possible comparison between values. For instance, how can we know if the EIL for conventional coffee and tractor application is right on 13.34% or actually between 13 and 14% or even between 10 and 16%? How can we know if the EIL for conventional coffee (13.34%) is really lower (= statistically lower) than the EIL for organic coffee (14.48%)?
In this study, the classic and appropriate methodology was used to determine pest control decision-making indices. In all the works published to date on this subject, dispersion measures for these indices were not included. However, to meet this reviewer's suggestion, we inserted the confidence limit at 95% probability in the model determined in this work and presented in Figure 2A. Subsequently, we used the percentage values of these intervals to determine the confidence intervals for the decision-making indices. As a result, changes were made to the Abstract (lines 27 - 29), Material and Methods (lines 209 - 2012), Results (352 - 356), Figure 2A (lines 336 - 338), Table 5 (lines 363 - 364), Discussion (lines 440 - 450) and Conclusions (lines 474 - 476).
- Beware of significant numbers (this comment is also linked with uncertainty, see comment 1). Figures expressed throughout the manuscript are expressed with too many digits. I doubt that the authors can say with certainty that the EIL for conventional coffee and tractor application is right on 13.34%. I believe that 13.3% is already more plausible, and that 13% might actually be accurate enough with this experimental design and for practical recommendations.
Due to the previous suggestion, changes were made to the EIL and ET values throughout the work.
- Crop yield as a function of pest attack intensity. Looking at figure 2a, I do not see a quadratic function as the best fit for regression. Almost all of the data points are below the quadratic curve. I am convinced that a linear regression would be a better fit. Furthermore, figure 2b does not match with figure 2a. In figure 2b, the residues are presented are equally distributed between positive values and negative values, but in figure 2a, almost all the data points are below the quadratic curve and therefore almost all of the residues are negative. The distance between the curve and the data points reaches at least 40% in some case, far above the claim 27.03%.
This issue with the regression is all the more important that the results of this curve are then vastly extrapolated. For instance, the authors claim that there is less than 1% damage at 5% of pest intensity, and less than 3% at incidence of 10%, but there is no evidence of that. This claim is only based on the regression, not on data points (the lowest incidence recorded on farm is only a single point around 10% and with damages of around 20%). The authors also rely on the shape of the curve to claim that coffee is tolerant to L. coffeella. And this curve is of primary importance to calculate the indices EIL and ET.
3281 regression models were tested including the linear model and the results obtained were worse than the quadratic model in terms of probability (p), coefficient of determination (R2), biological significance of both the curve and the values of the calculated indices. Due to these reasons the quadratic model was selected.
- Equations and coefficients used in section 2.4 on economic thresholds. I was not familiar with the indices EIL and ET before the review of this article, but a rapid research helped me understand the concepts. Based on my understanding, I do not believe that the equations 4 to 7 are correct to calculate EIL and ET. In the current version of the manuscript, I do not understand how the authors proceed with calculations, and where they get some of their coefficients.
Equation 4: by definition, EIL is related to both economic losses from pest damages (L) and costs for pest control (C). Both terms must appear in equation 4, but this is currently not the case. Equation 4 should look like EIL = min(infestation) with costs from pest (L) = cost from treatment (C). Where do coefficients 0.029 and 0.5 come from?
Equation 5: economic losses (L) by the pest are not related to control costs (C), they are related to losses in yield (Y). Equation 5 should look like losses = CP*V0*(1-Y). Why is V0 taken as 40 bags/ha? On figure 2, it looks like most farms with pest incidence < 20% are on average around 70% of maximum yield (82 bags/ha), hence V0 ~ 57 bags/ha.
Moreover, the cost of control is defined in section 2.2. If efficacy of the measures are only 80%, this must be translated at residual damage from pest. In that case, we could have something like economic losses after treatment = 20% * L.
All equations contained in this work are correct! The equations were described in an order and format to facilitate readers' understanding. Equation (5) is classic for determining economic injury levels and was proposed by the group of professor Dr. Larry P. Pedigo from Iowa State University, considered the world's greatest authority of all time on Integrated Pest Management Programs. In this work, some classic works by Dr. Pedigo's team are cited. Equation (4) is specific to the relationship between the attack intensity of Leucoptera coffeella and coffee plants. This equation was determined in this work and is presented in Figure 2A. In this case, the variable to be calculated is the EIL (pest attack intensity) as a function of losses in plant yield. Therefore, it was presented in this way to enable the calculation of the desired index, that is, the EIL. Equation (6) is very simple and it calculates the production value of any crop. The value of production is obtained by the product of the crop yield (bags of coffee per hectare) and the value of the product (US$ per bag). Equation (7) determines the economic threshold, which is an index at which control measures must be taken so that the pest population does not reach the economic injury level.
Regarding the value of the coefficient k (efficiency of the control method). This index is an essential part of the classic equation (equation 5) proposed by Dr. Larry P. Pedigo and that is why it was used. As for the value of k = 0.8 (80% efficiency), this was done because this is the requirement in Brazil for an insecticide to be registered in Brazil to be used to control pests. It is worth noting that Brazil is one of the few countries in the world that makes this requirement. It is important to highlight that chemical control is a curative method and complementary to natural methods such as natural enemies, climatic elements, plant resistance and physiological disorders during the pests' life cycle. According to the work developed by Pereira et al. (2007a and 2007b) these factors are responsible for mortalities of 89% to 94.3% of Leucoptera coffeella. Therefore, an efficiency of 80% is adequate and it promotes adequate pest control.
Finally, ET is arbitrarily set at 75% of EIL, meaning that actions are to be taken “25% before the EIL threshold”. That’s ok, but the authors should explain this in the text, to help the reader understand. As it stands, it can appear to the reader that ET and EIL are calculated separately. While in fact, only EIL is calculated based on empirical evidence, and ET is simply set at 75% afterwards as a prevention measure.
Finally, ET is arbitrarily set at 75% of EIL, meaning that actions are to be taken “25% before the EIL threshold”. That’s ok, but the authors should explain this in the text, to help the reader understand. As it stands, it can appear to the reader that ET and EIL are calculated separately. While in fact, only EIL is calculated based on empirical evidence, and ET is simply set at 75% afterwards as a prevention measure.
A justification for using this value was inserted in the Material and Methods (lines 249 -268) to meet the reviewer's suggestion.
Introduction
The introduction is clear, well written and backed with good references. There are a few minor comments to improve it:
- 29: coffee is not the 2ndmost traded commodity. Oil, iron, gold and many more are above coffee.
Phrase removed! Thanks for the information.
- 36: FAOstat (= reference [2]) does not give figures about coffee growers/consumers.
Removed.
- 51 and definition of EIL: if I am not mistaken, EIL is defined as the lowest density of pest at which economic damages match the costs of control measures. Please check the definition.
The definition of EIL was changed.
- 76: the previous indices based on expert knowledge and empirical evidences (20 to 30%) are key to this article. They must be compared to the values found by the authors and discussed later on in the article.
Add in the discussion.
Material and Methods
- Please provide more information on the farm area: how were they selected, what is the climate there (especially since the authors mentioned climate as an important factor for damages in the introduction), what is the average productivity, and so on.
As described in Material and Methods, seven central pivots with coffee plantations were evaluated for five years. Each pivot owned 100 hectares. The variation in yield of coffee crops in the central pivots during the five years is shown in Figure 2A. In Figure 2, each circle represents the yield of coffee plants in a pivot over the five years of conducting this study.
- 83: I was not familiar with central pivots. I suggest to replace with “central pivot irrigation systems”, which is more easily understandable, at least the first time that you mention central pivots.
Done!
- 2 Pest control costs: the authors repeat several times “research was carried out” but do not explain how it was carried out. What was the method of data collection (interviews, online shops websites, existing guidelines, literature review, ...)? How were the sources of information selected, and how many replicates were used to confirm the information? What were the variables investigated (cost per product, amount of products to use, possible methods of applications,…)?
As reported in Material and Methods, the first selection criterion for insecticides to be used in the calculations is the fact that the products are registered with the Ministry of Agriculture in Brazil for controlling Leucoptera coffeella in coffee crops. The second criterion was the products most used in coffee plantations to control this pest and for this purpose, consultations were made with farmers and technicians who work in this area. In research on product prices and application technologies, consultations were made with specialized stores and companies. This is reported in the Material and Methods (lines 214 - 221).
- Table 1 does not seem necessary. Instead, I suggest to describe the possible technologies directly in the text. This was done in l.185-188 and l.213.215. These sections would be better suited in the material and methods.
Table 1 removed.
- 3 Determination of crop yield curve. The amount of work necessary to collect this dataset over 5 years must be huge and provides solid ground for this research.
Thanks.
- Equation 3. Yield is a measure expressed in kg or bags per unit area, not in percentage. If the authors want to express it in percentage, they must name it differently. Furthermore, where does MPd = 82 bags/ha come from?
Equation (3) was used to transform productivity (bags/ha) into yield (%). This was done due to the fact that when determining economic injury levels (formula 5), crop yields are used as a percentage (lines 225 - 229).
Results
- The current presentation of the conventional and organic control costs is clear and can be kept as such. Nonetheless, I want to suggest an alternative and more concise representation of these costs. In my opinion, table 2 could aggregate the costs of insecticides for conventional treatments (top section of current table 2) and organic treatments (top section of current table 3). Table 3 could list the total cost of each treatment in a format like:
|
|
Avg cost of Insecticide (US/ha) |
Application method |
Cost of application (US/ha) |
# of application |
Total cost of application (US/ha) |
Total cost (US/ha) |
|
Conventional |
123.5 |
Manual |
30.7 |
5 |
153.5 |
277 |
|
conventional |
123.5 |
Tractor |
17.4 |
5 |
87 |
… |
|
organic |
32.2 |
Manual |
30.7 |
6 |
184.2 |
… |
|
|
|
|
|
|
|
|
We created this table according to the suggestion. However, in the end this table was very large and did not facilitate interpretation. Therefore, we decided to maintain the original representations.
- In addition, I suggest the authors to add a graphical representation of EIL and ET, at least for one of the treatment. This could be a line representing cost from treatment, a line representing the cost from pest damage. EIL will be at the intersection of these lines. This would help reader to understand the equations used in the manuscript and the meaning of these decision-support indices.
The costs are described in detail in Tables 2 to 4. The total costs, the economic injury levels and economic thresholds are presented in Table 5. Such as the costs (US$/ha) and the economic injury levels and economic thresholds ( percentage of leaves mined by Leucoptera coffeella) have different units, it is not possible for there to be an intersection between these two groups of variables.
Discussion
- The current discussion section is very pragmatic-oriented, and discussed the use of different application methods depending on farmers’ context. It is sufficient as it is, if the main goal of the authors is to provide decision-support indices for farmers. On the other hand, from a scientific point of view, I would suggest the authors to discuss the results in the larger context of integrated pest management. For instance, it has been shown with coffee leaf rust disease, that damages in year 1 result in yield losses in year 1, but also year 2 and 3 (see Cerda, R., Avelino, J., Gary, C., Tixier, P., Lechevallier, E., & Allinne, C. (2017). Primary and Secondary Yield Losses Caused by Pests and Diseases: Assessment and Modeling in Coffee. PLoS ONE, 12(1), e0169133. doi:10.1371/journal.pone.0169133). The authors could also quickly discuss the environmental and sanitary impacts from the different treatment methods.
Changes were made to the Discussion to meet the reviewers' suggestions. Thus, texts were inserted, modified or removed in the Discussion. In relation to the work of Cerda et al. (2014) it was conducted under controlled conditions in an experimental station, a small area (1440 m2) with 720 coffee plants for three years. The intensity of attack by Leucoptera coffeella was very low (1%, 2% and 4% in the first, second and third year) which did not influence the yield of coffee plants. In this study, the biggest attack was from the fungus Cercospora coffeicola with 26%, 22% and 23% attacks in the first, second and third year. This study was conducted in commercial coffee plantations in the region where L. coffeella is located, that is, in the Brazilian cerrado region. This study was conducted for five years in an area of 700 hectares. In commercial crops, which are the very reality of the studied phenomenon, there was a variation of up to 54.20% of mined leaves, which made it possible for the first time in the world to establish a highly significant regression curve (p < 0.0001) between the yield of coffee plants as a function of of the attack intensity of L. coffeella.
Conclusion
- Are the EIL and ET really different in organic and conventional management?
- If the goal is to come up with practical guidelines for farmers, shouldn’t the thresholds be simplified and rounded up or down?
Due to your suggestions, the EIL and EL were combined into a single value. As a result, the conclusion was changed (lines 474 - 476).
Reviewer 3 Report
Comments and Suggestions for Authors
Filho et al investigate and compare the cost of different pest management techniques in conventionally and organically grown coffee crops. Given the significance of the study, many opportunities are present to improve:
1- Currently, how the manuscript is written and structured is unsuitable for "plants" where readers are from diverse research fields. The abstract has to be more general which indicates the significance of the study. More context and explanation need to be given which is specific to the entomology field.
2- Due to the high interest in organic farming, the authors need to give a compelling story indicating why the current study was done comparing conventional with organic. Organic farming is environmentally friendly and keeps water pollution low. This needs to be highlighted in the introduction and discussion. Please cite appropriate papers. Some are mentioned here for assistance but authors can cite more papers as deemed fit e.g., https://link.springer.com/article/10.1007/s11356-021-15258-7; https://doi.org/10.1038/s41477-020-0656-9.
Comments on the Quality of English Language
English should be improved. Control indicates control for experiments. In the present manuscript, control should be replaced with pest management or something similar to reduce confusion.
Author Response
Reviewer 3
- Currently, how the manuscript is written and structured is unsuitable for "plants" where readers are from diverse research fields. The abstract has to be more general which indicates the significance of the study. More context and explanation need to be given which is specific to the entomology field.
Done!
- Due to the high interest in organic farming, the authors need to give a compelling story indicating why the current study was done comparing conventional with organic. Organic farming is environmentally friendly and keeps water pollution low. This needs to be highlighted in the introduction and discussion. Please cite appropriate papers. Some are mentioned here for assistance but authors can cite more papers as deemed fit e.g., https://link.springer.com/article/10.1007/s11356-021-15258-7; https://doi.org/10.1038/s41477-020-0656-9.
Done!